

# A *docked* mutation phenocopies *dumpy* oblique alleles *via* altered vesicle trafficking

Suresh Kandasamy, Kiley Couto and Justin Thackeray

Department of Biology, Clark University, Worcester, Massachusetts, United States

## ABSTRACT

The Drosophila extracellular matrix protein Dumpy (Dpy) is one of the largest proteins encoded by any animal. One class of *dpy* mutations produces a characteristic shortening of the wing blade known as oblique ($dpy^o$), due to altered tension in the developing wing. We describe here the characterization of *docked (doc)*, a gene originally named because of an allele producing a truncated wing. We show that *doc* corresponds to the gene model *CG5484*, which encodes a homolog of the yeast protein Yif1 and plays a key role in ER to Golgi vesicle transport. Genetic analysis is consistent with a similar role for Doc in vesicle trafficking: *docked* alleles interact not only with genes encoding the COPII core proteins sec23 and sec13, but also with the SNARE proteins synaptobrevin and syntaxin. Further, we demonstrate that the strong similarity between the $doc^1$ and $dpy^o$ wing phenotypes reflects a functional connection between the two genes; we found that various *dpy* alleles are sensitive to changes in dosage of genes encoding other vesicle transport components such as *sec13* and *sar1*. Doc's effects on trafficking are not limited to Dpy; for example, reduced *doc* dosage disturbed Notch pathway signaling during wing blade and vein development. These results suggest a model in which the oblique wing phenotype in $doc^1$ results from reduced transport of wild-type Dumpy protein; by extension, an additional implication is that the $dpy^o$ alleles can themselves be explained as hypomorphs.

## INTRODUCTION

About one-third of all eukaryotic proteins pass through the endoplasmic reticulum (ER) on their way to a variety of ultimate destinations. Anterograde traffic from the ER to the Golgi apparatus is highly conserved across all eukaryotes and depends on coat protein complex II (COPII) vesicles, while retrograde transport from the Golgi back to the ER relies on COPI-coated vesicles (*Lord, Ferro-Novick & Miller, 2013*; *Venditti, Wilson & De Matteis, 2014*). COPII vesicles are assembled in a step-wise manner, beginning with activation of the small G-protein Sar1. This leads to recruitment of a Sec23/Sec24 heterodimer, followed by the assembly of an outer coat heterotetramer constructed from Sec13/Sec31 molecules. However, it is clear that this description is incomplete for some cargoes; for example, traffic of collagen requires additional factors (TANGO1 and cTAGE5) to interact with Sec23/24, presumably to allow such a large protein to be packaged inside the vesicle (*Malhotra & Erlmann, 2011*). Upon reaching the Golgi the

Corresponding author
Justin Thackeray,
jthackeray@clarku.edu

vesicle is tethered by a combination of a multi-protein complex known as TRAPP, a Rab family member called Ypt and several other Ypt-interacting proteins. Finally, fusion of the vesicle to its target membrane depends on a set of proteins known as SNAREs (soluble N-ethylmaleimide-sensitive factor attachment protein receptor) (*Lord, Ferro-Novick & Miller, 2013*).

A particularly challenging cargo in Drosophila is presented by Dumpy (Dpy), with some isoforms being 23,000 amino acids long and estimated at 2.5 MD (*Wilkin et al., 2000*). Dpy is probably the largest protein made by Drosophila and is among the largest proteins made by any organism; it was recently shown that Dpy is trafficked *via* a bulky cargo-specific pathway that depends on the Drosophila Tango homolog, Tango1 (*Rios-Barrera et al., 2017*). Dpy is membrane-anchored but the vast majority of the molecule projects into the extracellular matrix (ECM), where it plays a role in attachment between epithelial cells and adjacent tissues (*Wilkin et al., 2000*). Loss of *dpy* function is lethal, but its function can be observed from loss of function restricted to the wing, which results in blisters due to separation of the apical ECM (aECM) from the epidermis (*Bokel, Prokop & Brown, 2005*; *Prout et al., 1997*). Much of the Dpy protein consists of many copies of three repeats; in addition to over 300 EGF (Epidermal Growth Factor) repeats there are two distinct repeats unique to Dumpy-185 copies of a Dumpy-specific motif referred to as DPY, and about 40 copies of a sequence known as PIGSFEAST (*Wilkin et al., 2000*). Some of the EGF repeats are substrates for Eogt, an extracellular-specific O-Glc-NAc (O-linked N-acetylglucosamine) transferase, and this modification affects Dpy function in the aECM (*Sakaidani et al., 2011*). *dumpy* (*dpy*) has been the subject of intensive genetic study over many decades, due to the combination of several easily identified mutant phenotypes and the huge mutational target of the gene itself. The locus is genetically complex, with alleles falling into three main classes: oblique ($dpy^o$), vortex ($dpy^v$) and lethal ($dpy^l$), and many showing more than one of the three phenotypes (*Carlson, 1959*). The oblique class gave the gene its name, producing a short and distally clipped wing, with a reduced number of cells in the wing and an indentation close to the margin between veins L3 and L4 (*Waddington, 1940*); this is due to abnormal tension between cells within the developing wing, which skews its final shape (*Etournay et al., 2015*).

The Drosophila wing is a large and completely dispensable organ, with a highly stereotyped pattern of veins. In combination with an array of genetic tools–particularly a collection of wing-specific *GAL4* drivers–these features have allowed studies of the fly wing to greatly expand our understanding of organogenesis and patterning (*Baker, 2007*; *Neto-Silva, Wells & Johnston, 2009*). The Drosophila wing develops from a group of about 20 cells in the embryo into an imaginal disc of about 75,000 cells by the end of the third larval instar. During pupariation the wing disc everts and cuticle is secreted to form the dorsal and ventral surfaces, which are held together by a meshwork of ECM molecules. The vesicular trafficking machinery plays a central role at all stages in this process, being necessary for not only for placement of membrane-bound receptors such as Notch, Frizzled, Patched, Thickveins, Punt and EGFR, but also for secretion of various

morphogens including Wingless (Wg), Hedgehog (Hh), Decapentaplegic (Dpp) and Vein (Vn).

During a screen to identify new alleles of the Drosophila gene *amontillado (amon)*, a lethal complementation group distinct from *amon* was also identified in an adjacent genomic region. Among the mutations recovered was a single viable allele, producing an indentation in the distal part of the wing; the new gene was named *docked* because of this phenotype (*Rayburn et al., 2003*). The lethal alleles of *doc* (*doc¹³, doc¹⁶ and doc⁹⁹*) fail to complement each other, but are viable when heterozygous with the viable allele *doc¹*, although they all fail to complement the wing phenotype of *doc¹* (*Rayburn et al., 2003*). Noting the similarity between the *doc¹* and *dpyᵒ* wing phenotype we began investigating the *doc* alleles, to reveal whether Doc may participate with Dpy during wing development. We describe here our discovery that *doc* encodes the Drosophila homolog of yeast YIF1 (Yip1-interacting factor), and this finding led us to the discovery that the *dumpy* oblique phenotype can be explained by aberrant trafficking of wild-type Dpy.

## MATERIALS AND METHODS

### Fly stocks

Flies were raised on a standard cornmeal, molasses, yeast and agar medium at 25°C. *doc¹/TM3Sb, doc¹⁶/TM3 Sb, Ser and doc⁹⁹/TM6Tb* were kind gifts from Dr. Michael Bender (Univ. of Georgia, Athens, Georgia). A strain carrying the *UAS-dNSF2^(E/Q)* construct (*Stewart et al., 2001*) was kindly provided by Hugo Bellen (Baylor College of Medicine, Houston, TX, USA). A *UAS-RNAi-doc* strain (ID#2679) was obtained from the Vienna Drosophila Research Center. *dpy^(DvlG5)* was isolated from an EMS mutagenesis screen (T. Abbeyquaye and J. Thackeray, unpublished). All other stocks were obtained from the Bloomington Drosophila Stock Center at Indiana University.

### Generating a wild type *CG5484* transgenic rescue construct

DNA was extracted from *wild-type* canton S strain flies and PCR was carried out using Native Pfu polymerase (Stratagene Inc, La Jolla, CA, USA) according to the manufacturer's instructions. The *CG5484* transcription unit and about 1 kb of intergenic sequence on either side were amplified in two overlapping PCR products and joined using a common *Nsi*I cut site to generate a full-length wild-type *CG5484* rescue construct. The forward and reverse primers for the more 5′ product were 5′TTCGCGCTGGCTCAAACTGCCTA3′ and 5′-CCAAGAGC**T**CCCAGTGCAACCA-3′ respectively; the underlined base is a mismatch used to incorporate a SacI site into the product. The 3′ half of *CG5484* was amplified using the primer pair 5′GCGAAGTAGTACTT**A**AGCTTGGC3′ and 5′-GGATGGATTCATCTA**G**ACCTCGG-3′ respectively; the underlined base in the former is a mismatch used to incorporate a *Hin*dIII site, in the latter primer the underlined base introduces an *Xba*I restriction site. A 4.1 kb *Sac*II/*Xba*I fragment that begins 1124 bp 5′ of the predicted translation start and ends 1,352 bp 3′ of the predicted stop codon was ligated into the pCaSpeR4 germline transformation vector. The genomic insert in this vector was verified by sequencing and then injected into *white¹¹¹⁸* embryos to produce

transgenic lines expressing wild-type *CG5484*; transgenic injection was performed by Genetic Services Inc, Sudbury, MA, USA.

## RESULTS

### Alleles of *doc* interact genetically with *dumpy* mutations

We noticed a strong similarity in wing phenotype between $doc^1$ homozygotes and the oblique class of *dumpy* (*dpy*) mutants, which suggested to us that Doc and Dpy may be involved in a common pathway during wing development. We outcrossed four *doc* alleles and confirmed their allelic status, the lethality of $doc^{13,16,99}$ and the oblique wing phenotype of $doc^1$ (Fig. 1). These initial experiments also revealed that $doc^1$ is semi-dominant: $doc^1$/+ heterozygotes show a very slight indentation in the distal part of the wing. Next we looked for genetic interaction between *doc* and *dpy*. We found that the oblique wing phenotype of two dominant *dpy* alleles, $dpy^D$ and $dpy^{DvlG5}$ as well as a recessive allele $dpy^{olvr}$ is strongly enhanced when placed in a background heterozygous for $doc^1$, producing a significant further reduction in wing length (Fig. 1). This interaction is consistent with Doc and Dpy playing roles in a common signaling pathway. However, two lethal alleles of *doc*, $doc^{16}$ and $doc^{99}$, did not interact with any of these *dpy* alleles (data not shown), suggesting that the $doc^1$ mutation is qualitatively different from the other two *doc* alleles.

### Doc is encoded by the gene model *CG5484*

The *doc* alleles were identified originally by either their lethality ($doc^{13,16,19}$) or their wing phenotype ($doc^1$) when heterozygous with two overlapping third chromosome deficiency chromosomes, *Df(3R)ro80b* and *Df(3R)Tl-X* (Rayburn et al., 2003), implying that the *doc* transcription unit lies within the genomic interval common to both deficiencies. We amplified genomic DNA from the *doc* mutant chromosomes, using primers in coding regions from several of the approximately 20 genes within this interval. We targeted loci predicted to encode proteins that seemed most likely to play a role in regulating signaling pathways in wing development: *CG6420*, *CG5484*, *beat VII* and *scribbled*. Nucleotide sequences from these candidate genes in various $doc^1$, $doc^{16}$/TM3Sb and $doc^{99}$/TM6Sb backgrounds were compared to wild-type; we did not observe unique mutations in any of the candidate genes except one: *CG5484*. The $doc^{16}$ chromosome carries a nonsense mutation in the predicted *CG5484* open reading frame, affecting amino acid position 162 and therefore eliminating more than half of the predicted full length protein (Fig. 2A). However, we were unable to identify mutations in *CG5484* in either the $doc^1$ or $doc^{99}$ chromosomes. To establish whether *CG5484* corresponds to *doc*, we generated a wild type genomic construct containing all predicted *CG5484* exons and introns, as well as approximately 1 kb of additional 5′ and 3′ sequence beyond the predicted transcription unit. A single copy of this transgene completely rescued both the oblique wing phenotype of $doc^1$ homozygotes and the lethality of $doc^{16}$ (Figs. 2B–2D). The transgenic construct did not rescue the lethality of $doc^{99}$, but this is likely due to a tightly linked lethal mutation in another locus, because the *CG5484* construct was able to rescue the lethality of $doc^{16}$/$doc^{99}$ transheterozygotes (data not shown). Taken together

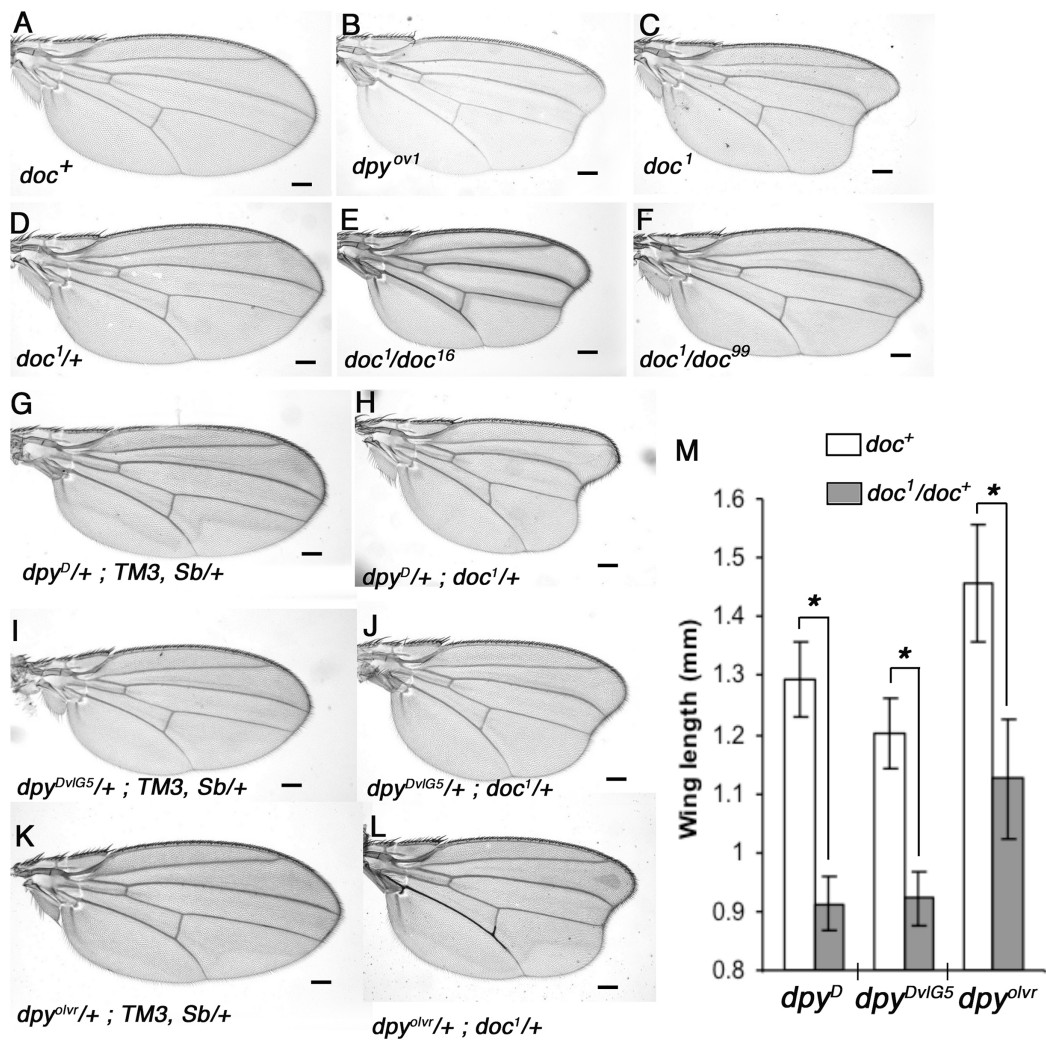

**Figure 1 Allelic nature of *doc* mutations and interaction of *doc¹* with *dumpy* oblique alleles.**
(A) Wildtype wing. (B–C) Wings of *doc¹* and *dpy^ov1* showing the oblique wing phenotype. (D) Slight oblique wing phenotype in *doc¹* heterozygote. (E–F) Transheterozygous combinations of *doc¹* with *doc¹⁶* and *doc⁹⁹* show the allelic nature of the three mutations. (G–L) Wings of *doc¹* with three different *dumpy* alleles show a strong interaction between the two genes. Representative wings from adults of the genotypes indicated; scale bar represents 100 um. (M) Wing lengths of genotypes shown in panels G–L; error bars indicate standard error of the mean (n = 25 for G–J, n = 22 for K, n = 17 for L). Length was measured as a straight line from the intersection of the anterior crossvein with vein L3, to the intersection of L3 with the margin, using ImageJ. Asterisk indicates significant differences in wing lengths between *doc⁺* and *doc¹/doc⁺* siblings in each of the three *dpy* backgrounds; two-sample t test assuming unequal variances, *dpy^D* (df = 67) t = 27.35, $p = 2.2 \times 10^{-38}$; *dpy^{DvlG5}* (df = 89) t = 19.55, $p = 2.87 \times 10^{-34}$; *dp^{olvr}* (df = 21) t = 10.66, $p = 3.13 \times 10^{-10}$.

with the nonsense mutation in the *doc¹⁶* chromosome, these germline rescue experiments demonstrate that *CG5484* encodes Doc. Henceforth we refer to *CG5484* as *docked*.

## *doc* encodes a Yif1 homolog

The FlyBase annotation of *doc* shows that the gene produces three slightly different mRNAs *via* alternative splice site usage at the 3′ end of exon one

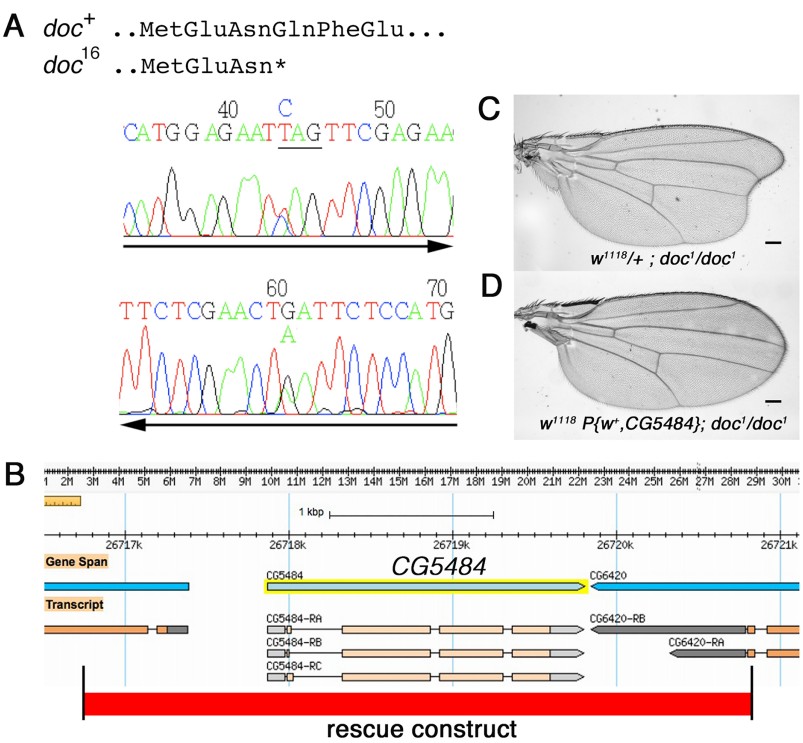

**Figure 2 The *doc* complementation group corresponds to the gene model *CG5484*.** (A) Genomic DNA sequencing chromatograms of *doc*[16] heterozygotes, showing that the same transition mutation is present (as a double peak) on both strands, producing a Q162STOP nonsense mutation. The affected glutamine codon is underlined in the upper chromatogram. (B) The genomic region around *CG5484*; the 4.1 kb of DNA represented in the germline rescue construct is indicated by the red-shaded rectangle beneath the map. (C) Representative wing from *w*[1118]; *doc*[1] homozygote. (D) Representative wing from *w*[1118]; *doc*[1] homozygote carrying one copy of the *P{w+, CG5484}* construct shown in (B); the oblique wing phenotype seen in (C) is completely rescued in panel D. The animals in (C and D) are siblings; scale bars represent 100 um.

(*McQuilton, Pierre & Thurmond, 2011*), predicting a set of three very similar Doc isoforms: A (397aa), B (393aa) and C (402aa). Genbank searches using the longest isoform revealed that Doc has no other homologs in Drosophila (not shown). However, these searches also clearly demonstrated that Doc is a member of the YIP (Ypt-interacting protein) superfamily that was first identified in the budding yeast, *S. cerevisiae* (*Matern et al., 2000*). Comparison of Doc with the most similar YIP family members from yeast (YIF1) and human (Yif1A) shows relatively poor conservation over the N-terminal third of Doc, but the region homologous to the more C-terminal two thirds shows substantial conservation among all three species (Fig. S1). A characteristic feature of YIP family members is the presence of five likely membrane-spanning domains; a hydropathy plot of the Doc sequence reveals that these are conserved within Doc (Fig. S2).

Doc shows greatest similarity to yeast Yif1 (Yip1 interacting factor), which is an integral membrane protein required for membrane fusion of ER-derived vesicles (*Barrowmann et al., 2003*) and is essential for ER to Golgi vesicle-mediated transport in *S. cerevisiae* (*Heidtman et al., 2005*). While we were preparing this manuscript others reported the same finding, that *CG5484* is a Yif1 homolog in Drosophila (*Wang, Wang & Yu, 2018*).
We used PCR amplification from mRNA to confirm that the A, B and C splice forms are generated in wild-type adult tissue; these experiments also suggest that the B form is the most abundant of the three (Fig. S3). There are no reports of alternative splicing in the exons encoding the N-terminal end of the yeast or human YIF1 homologs; so we examined other insect species to determine whether the three alternative splice forms are evolutionarily conserved. All eight Drosophila species we examined show conservation of the three alternative splice donor (GT) sequences at identical locations in their respective *doc* homologs, suggesting that the three Doc isoforms are conserved in the other species and therefore likely to have distinct functional roles (Fig S3). In summary, the Doc sequence strongly suggests it is a YIF1 homolog and is therefore very likely to be involved in vesicle trafficking in Drosophila.

## Interaction of *doc* with genes encoding COPII core proteins

*S. cerevisiae yif1* mutants exhibit defective COPII vesicle transport (*Matern et al., 2000*) and its Yif1 protein product has been described as either a COPII vesicle component, or a resident of the Golgi necessary for ER-derived COPII vesicle docking (*Heidtman et al., 2003*; *Jin et al., 2005*; *Matern et al., 2000*). To determine whether Doc plays a similar role in *Drosophila* we looked for genetic interaction between *doc* and the genes encoding some of the core components of COPII vesicles. The COPII coat assembles by a stepwise deposition of Sar1-GTP, Sec23–Sec24, and Sec13–Sec31 onto ER exit sites (*Gurkan et al., 2006*). Sec23–Sec24 and Sec13–Sec31 are heteromeric protein complexes that are major constituents of the inner and outer layers of the COPII coat (*Matsuoka et al., 1998*). We found a strong interaction between *doc* and *sec23* mutations: there is a noticeable enhancement of the oblique wing phenotype in $doc^1 sec23^{j13C8}$ double heterozygotes (Fig. 3), whereas the double heterozygous combination of $doc^1$ and $sec23^{9G}$ alleles was lethal. Similarly, in a background heterozygous for $sec13^{01031}$ the oblique wing phenotype of $doc^1$ heterozygotes was markedly more severe compared to $doc^1/+$ alone (Fig. 3). In the case of *sar1* we observed no change to the severity of the wing in $doc^1/sar1^{05712}$ flies, but this combination showed some evidence of semi-lethality: the number of $doc^1$ +/+ $sar1^{05712}$ double heterozygotes was reduced by about 40% compared to the combined number of $doc^1/TM3$ and $sar1^{05712}/TM3$ siblings (six *vs* 30 respectively). These synergistic interactions strongly suggest a role for Doc in COPII vesicle trafficking. We did not observe any interaction in animals that were doubly heterozygous for either $doc^{16}$ or $doc^{99}$ and also $sec23^{j13C8}$, $sec13^{01031}$ or $sar1^{05712}$.

## Doc has roles in vesicle fusion

Studies from yeast have revealed that Yif1 forms a complex with Yip1 and interacts with Rab GTPases in the acceptor Golgi membrane, suggesting a role for Yif1 in tethering of COPII vesicles (*Matern et al., 2000*). Accurate matching of vesicles to the correct target depends on v-SNARE proteins in the COPII membrane forming a complex with specific t-SNAREs in the target membrane. We therefore generated heterozygous combinations of *doc* mutations with mutations of genes encoding either the v-SNARE component Synaptobrevin (*syb*), or one of the t-SNARE proteins Syntaxin (*Syx*), using the

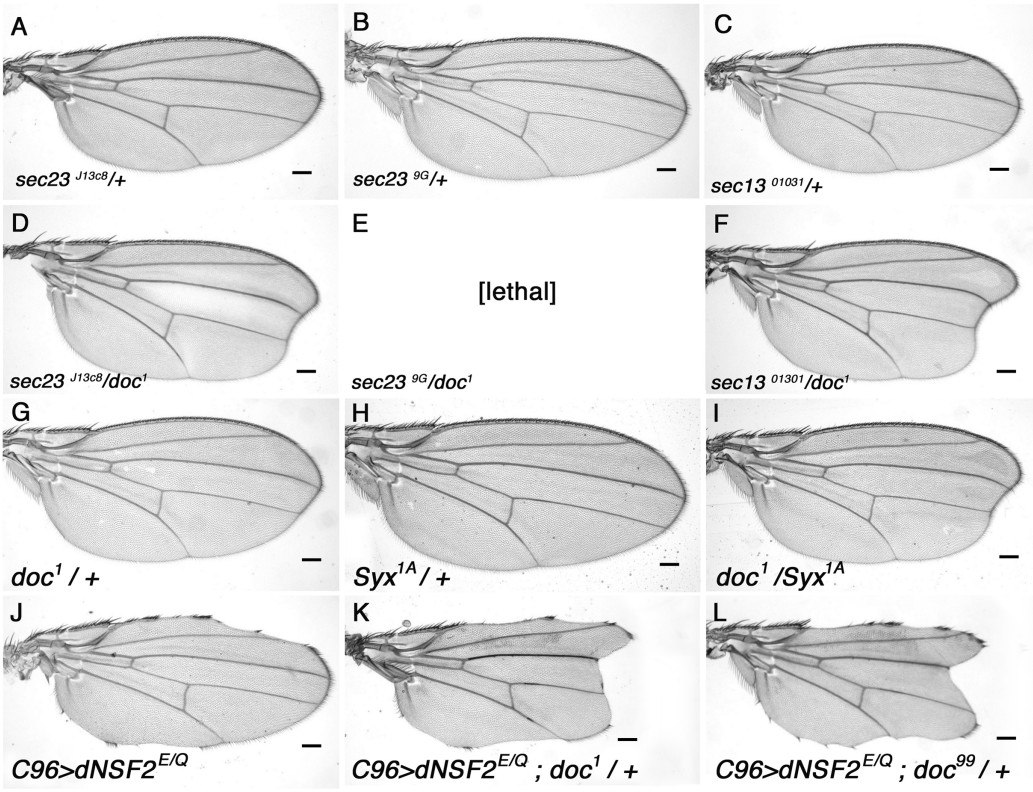

**Figure 3 The *doc* alleles interact strongly with genes involved in vesicle trafficking.** (A–F) Strong enhancement of the oblique wing phenotype in *doc¹* heterozygotes when also heterozygous for mutations in *sec23* and *sec13*. (G–I) Enhanced wing phenotype of *doc¹* in a *Syx1A* heterozygous background. (J–L) Wing nicks in *UAS-dNSF2^{E/Q};C96-GAL4* wings are enhanced in backgrounds that are also heterozygous for *doc¹* and *doc⁹⁹*. The same result was found for *doc¹⁶* (not shown). Representative wings from adults of the genotypes indicated; scale bars represent 100 um.

syntaxin 1 homolog encoded by *Syx1A*. We observed clear interaction between *doc¹* and *Syx1A*, first, by a modest enhancement of the *doc¹* oblique wing phenotype in *doc¹* +/+ *Syx1A^{Δ229}* flies (Figs. 3G–3I) and second, because the *doc⁹⁹* mutation showed semi-lethality even in heterozygous condition if the dosage of Syx1A was also reduced using the null allele *Syx1A^{Δ229}*: the cross produced only 36% of the number of *doc⁹⁹* +/+ *Syx1A^{Δ229}* adults compared to their *doc⁹⁹*/TM3 sibs (five *doc⁹⁹* +/+ *Syx1A^{Δ229}* adults compared to 28 *Syx1A^{Δ229}*/TM3 or *doc⁹⁹*/TM3). By contrast, we found no interaction between any *doc* allele tested and *syb* (data not shown). These results are consistent with a role for Doc in COPII vesicle fusion with the ER, and suggest a more critical involvement with events occurring on the target membrane.

Next, we investigated whether Doc might play a role further downstream of *cis*-membrane fusion. One aspect of this is the need for *cis*-residing protein complexes to be broken apart to be recycled for future *trans*-complex formation, maintaining anterograde vesicle transport. NSF is an ATPase involved in dissociation of SNAREs (*Banerjee et al., 1996*) and thus NSF may permit recycling of SNAREs to maintain a steady rate of protein transport. We perturbed dNSF function in the wing margin by expressing

the dominant negative $UAS\text{-}dNSF2^{E/Q}$ construct (*Stewart et al., 2001*) under the control of *c96-GAL4*, which drives expression in the wing margin (*Gustafson & Boulianne, 1996*), generating nicks. We found that this wing nick phenotype is enhanced in backgrounds that are heterozygous for any of $doc^1$, $doc^{16}$ or $doc^{99}$ (Figs. 3J–3K). The interaction of *doc* with *dNSF2* is consistent with a role for Doc in vesicle trafficking, and suggests a role for Doc that matches that of yeast Yif1 in vesicle fusion (*Barrowmann et al., 2003*).

### *doc* is required broadly during development

Flies homozygous for either the $doc^{16}$ or $doc^{99}$ alleles, as well as $doc^{16}/doc^{99}$ transheterozygotes, do not survive development, with a lethal phase during embryogenesis (data not shown). Similarly, we found that expression of a *UAS-doc-RNAi* construct using the broadly expressed *wg-GAL4* driver is lethal (data not shown). Furthermore, restricting the *doc* knockdown to trachea using the *breathless-GAL4* driver (*Matusek et al., 2006*; *Reichman-Fried & Shilo, 1995*) was also lethal, consistent with a general requirement for Doc during organogenesis. The adult wing provides a convenient way to examine cell-lethal functions in the adult, because it is completely dispensable. RNAi knockdown of *doc* in the wing using the dorsal compartment driver *MS1096-GAL4* resulted in complete loss of wing structures (not shown). However, restricting knockdown of *doc* to the dorsal-ventral (DV) wing boundary using the *C96-GAL4* driver (*Gustafson & Boulianne, 1996*) resulted in partial loss only of marginal tissue (Fig. 4).

### A role for Doc in Notch ligand transport

The wing nicks produced by targeted *doc* knockdown, as well as the synergistic interactions between *doc* alleles and mutations in genes involved in vesicle trafficking, suggests a role for Doc during export of developmentally important signaling ligands in the wing. Spatially restricted expression of the Notch (N) receptor and its ligands, Serrate (Ser) and Delta (Dl) play key roles in wing margin development (*Artavanis-Tsakonas, Rand & Lake, 1999*; *de Celis, Garcia-Bellido & Bray, 1996*; *Neumann & Cohen, 1996*; *Panin & Irvine, 1998*) and marginal wing nicks very similar to those we observed with *C96 > doc-RNAi* are observed in some alleles of both *N* and *Ser* (*Thomas, Speicher & Knust, 1991*; *van de Hoef, Bonner & Boulianne, 2013*). We therefore examined whether there are genetic interactions between the genes encoding these molecules and *docked*.

Wings of $Dl^7$ mutants have thickened veins and delta-like wing venation patterns and we found that this *Dl* venation phenotype is enhanced by all three *doc* alleles tested: $doc^1$, $doc^{16}$ and $doc^{99}$ (Figs. 4A–4C). This was manifested by much increased thickening of vein L2, as well as frequent thickening and/or delta formation at the distal ends of veins L3 and L4. A similar enhancement was observed when *doc* dosage was reduced in a background that was heterozygous for both $Dl^{RevF10}$ and $Ser^{RX82}$ (Figs. 4D–4F). We found a particularly strong interaction between *doc* and *N*. Halving the dosage of *docked* with either $doc^1$ or $doc^{16}$ was lethal in a $N^{nd-3}$ heterozygous background, while in a $doc^{99}$

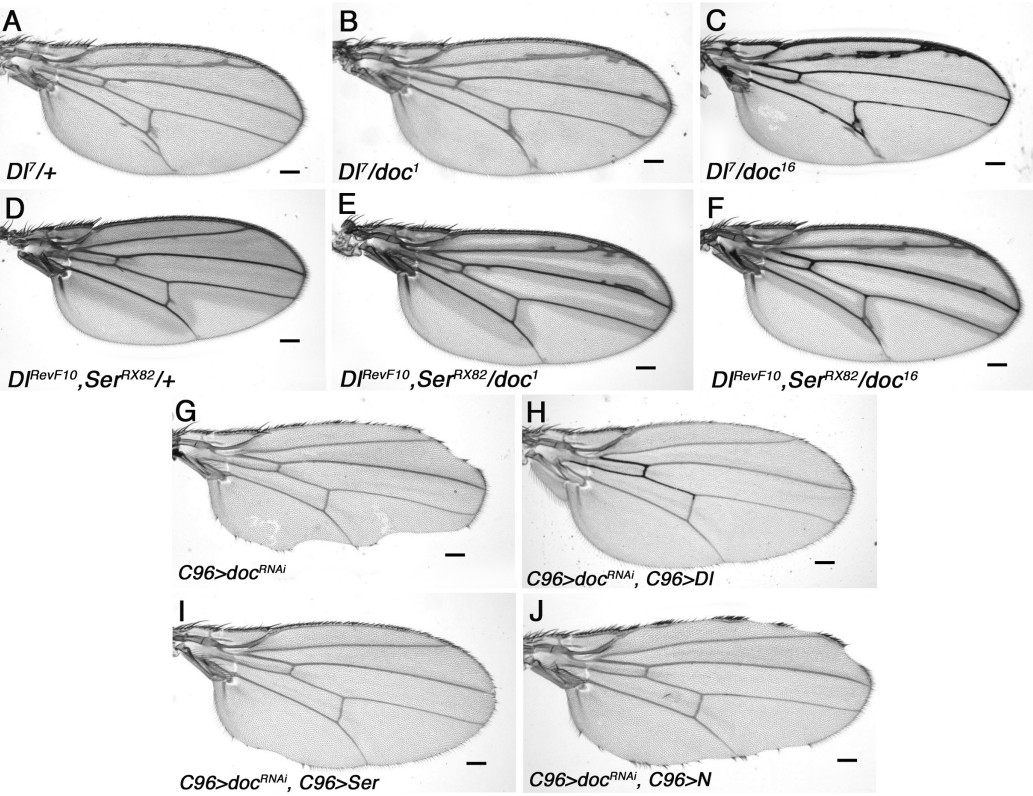

**Figure 4 Genetic interaction of *doc* alleles with members of the *Notch* pathway.** (A–F) Mild enhancement of the Delta wing venation phenotype in *doc1* and *doc16* heterozygous backgrounds. (G–J) Wing nicks of *UAS-Dicer2/+; UAS-RNAi-doc/+; C96-GAL4/+* are rescued by over-expression of *Delta* and *Serrate*, but not *Notch*. Representative wings from adults of the genotypes indicated; scale bars represent 100 um.

heterozygous background only an occasional $N^{nd-3}$/+; $doc^{99}$/+ escaper was recovered; these escapers did not have any observable change in wing phenotype (not shown). These results are consistent with Notch signaling being reliant on Doc-dependent vesicle trafficking.

Our finding that at least some Notch signaling depends on Doc suggests a possible cause of the wing nicks observed in the *C96 > RNAi-doc* animals: failed trafficking of N or its ligands. We tested this model directly, by overexpressing Dl or Ser, using *UAS-Dl* or *UAS-Ser* constructs driven by the *C96-GAL4* driver, in a background also expressing the *UAS-doc*-RNAi construct. We found that overexpression of Delta was sufficient to completely rescue the wing nicks resulting from RNAi knockdown of *doc*, whereas in the case of Serrate the rescue was strong, but not complete (Figs. 4G–4I). By contrast, overexpression of Notch using the same method did not rescue the *doc*-RNAi wing margin defect, although some degree of rescue was apparent in most animals (Fig. 4J). These findings strongly suggest that the wing nicks observed in *C96 > RNAi-doc* animals are due to reduced availability of Notch ligands, presumably as a result of lowered vesicle trafficking.

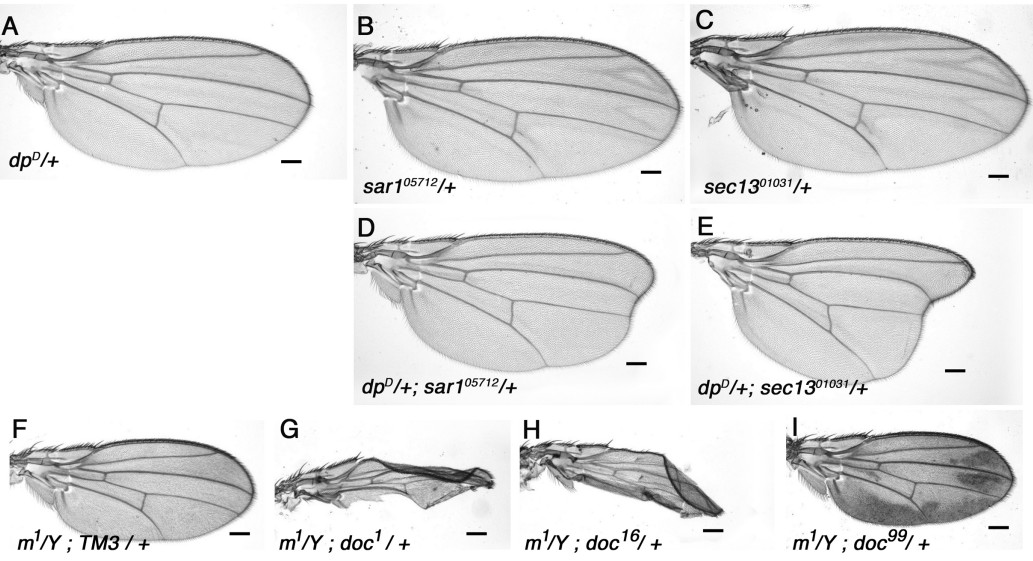

**Figure 5 Vesicle trafficking mutations strongly enhance a *dumpy* dominant oblique mutation.** (A) Wings of the dominant oblique wing mutant $dpy^D$ show a near-wildtype wing. (B, C) Wings of heterozygotes for either $sar1^{05712}$ or $sec13^{01031}$ are wild-type. (D, E) Animals simultaneously heterozygous for $dpy^D$ and either $sar1^{05712}$ or $sec13^{01031}$ have a strongly enhanced oblique wing phenotype. (F) Wing from a male $m^1$ hemizygote. (G–I) In backgrounds heterozygous for either $doc^1$ or $doc^{16}$, $m^1$ wings are crumpled, (I) whereas in $doc^{99}$ heterozygotes the wings show excessive pigmentation. Representative wings from adults of the genotypes indicated; scale bars represent 100 um.

**The interaction between *doc* and *dpy* is related to transport of Dpy**

Animals homozygous for $doc^1$ are a remarkably accurate phenocopy of the oblique class of *dpy* mutations, and we have shown in this study that a strong genetic interaction exists between the two genes (Fig. 1). Dpy resides in the extracellular matrix (*Wilkin et al., 2000*), so it can be assumed to pass through the vesicular trafficking pathway to reach its final destination. Given the similarity in their phenotypes, an obvious question is whether the $doc^1$ wing phenotype arises from reduced transport of Dpy. If this is true, then we might expect other mutations that disturb vesicle trafficking to worsen the severity of the oblique wing phenotype in *dumpy* mutants. We find that this is indeed the case: whereas $dpy^D$ heterozygotes show a barely detectable oblique phenotype, transheterozygous $dpy^D/+$; $sec13^{01031}/+$ and $dpy^D/+$; $sar1^{05712}/+$ animals both show a much more substantial oblique phenotype (Fig. 5). By contrast, no observable change in the severity of the oblique wing phenotype was observed in *trans*-heterozygous combinations of $dpy^D$ with either of two *sec23* alleles, $sec23^{9G}$ or $sec23^{j13C8}$ (data not shown). The interaction between $dpy^D$ and both $sec13^{01031}$ and $sar1^{05712}$ is consistent with a model in which the $doc^1$ oblique wing phenotype is due to an effect occurring *via* Dumpy.

The interaction of $dpy^D$ with alleles of *doc*, *sec13* and *sar1* all suggest a particular sensitivity of Dpy to reduced COPII vesicle transport, but we wondered if the same effect might be seen if vesicle trafficking at other stages was reduced. Jagunal (Jagn) is a conserved ER membrane protein required for replenishment of the ER membrane (*Lee & Cooley, 2007*), so we tested whether *jagn* mutants show any interaction with $dpy^D$.

Although there was no observable wing phenotype in adults (not shown), we found that the transheterozygous combination of $dpy^D/jagn^{G4067}$ is semi-lethal: only 29% (four $vs$ the expected 14) of the expected number of $dpy^D/jagn^{G4067}$ adults emerged compared to their $dpy^D/+$ or $jagn^{G4067}/+$ siblings. This result buttresses the idea that Dpy is highly sensitive to reduced COPII-dependent traffic, because Jagn is needed to maintain future rounds of COPII transport; it also shows that this effect is not specific to a particular stage of extracellular transport.

Halving the dosage of genes encoding components of the vesicle trafficking machinery would be expected to affect any protein that passes through the ER, and not just Dpy. We therefore looked for genetic interaction between $doc$ and other genes encoding proteins that are similar to Dpy in any of size, amino acid sequence or extracellular location. Dpy is one of a small group of proteins that contain a ZP (zona pellucida) domain, a motif that is unique to a subset of extracellular proteins (*Jovine et al., 2002*); all ZP domain-containing proteins can therefore be assumed to be trafficked between the ER and Golgi to reach their final destination. Sixteen Drosophila genes encoding proteins with a ZP domain are identifiable in release six of the Drosophila genome and we examined $P$-insertion alleles of ten for genetic interaction with $doc$. Nine of these ten–*dusky-like*, *Dusky*, *piopio*, *cypher*, *trynity*, *nyobe*, *nyeo*, *nomp-A* and *CG17111*–showed no interaction with any of $doc^1$, $doc^{16}$ or $doc^{99}$ when in various transheterozygous combinations (data not shown). However one member of this group, *miniature*, did show significant interaction with $doc$. Miniature is a transmembrane protein required for proper assembly of the chitin-based cuticle; adults homozygous or hemizygous for viable *miniature* ($m$) alleles have short wings (*Slatis & Willermet, 1954*). Wings of hemizygous $m^1$ males that were also heterozygous for either $doc^1$ or $doc^{16}$ had much more abnormal wings than $m^1/Y$ alone, being both narrow and misshapen (Figs. 5F–5H). Although the reduced size of the wings of $m^1/Y$; $doc^{99}/+$ males was typical of $m^1$ hemizygotes, there was a noticeable increase in the amount of darkly-pigmented tissue that is characteristic of some $m$ alleles (Fig. 5I). In addition, two transheterozygous combinations of $doc$ and $m$ were lethal: $m^{259-4}/+$; $doc^{16}/+$ and $m^{259-4}/+$; $doc^{99}/+$. By contrast, we found no interaction between $doc^1$ and $m^{259-4}$ (data not shown).

Finally, because Dpy is an exceptionally large protein known to use a distinct trafficking pathway for bulky cargoes (*Rios-Barrera et al., 2017*) we wondered whether the interaction between $doc^1$ and $dpy$ might also be related to the exceptional size of Dumpy. We were able to identify in Flybase only one other secreted Drosophila protein approaching its size: Mucin14A, which has a largest isoform of 16,233aa (*Syed et al., 2008*). We found no change in the phenotype of any of the three $doc$ alleles when they were placed in a background heterozygous for a viable allele of $Muc14A^{f00286}$, suggesting that the interaction of Doc with Dumpy is not solely due to its large size (not shown). Because the size of Dpy could conceivably be causing other Doc-dependent effects distinct from secretion, we also tested for interactions with genes encoding six relatively large intracellular proteins *sallimus* (longest predicted isoform given in Flybase (*St Pierre et al., 2014*) 18,488aa), *ankyrin2* (13,559aa), *muscle-specific protein 300* (13,540aa), *stretchin* (9,839aa), *bent* (8,933aa) and *shortstop* (5,501aa), but found no interaction when in

transheterozygous combinations with various *doc* alleles (not shown). These findings suggest that the interaction between $doc^1$ and *dpy* is unlikely to be due to the size of Dpy or the presence of a ZP-domain *per se*.

## DISCUSSION

We have shown here, by identification of a nonsense mutation in $doc^{16}$ and germline rescue of the mutant phenotypes in three independent *doc* alleles, that the gene model *CG5484* corresponds to *docked*. The predicted Doc amino acid sequence shows strong sequence similarity to yeast Yif1; it is also of similar size and contains the five predicted transmembrane domains that are typical of Yif1 homologs. Yeast Yif1 is necessary for COPII vesicle transport (*Matern et al., 2000*) and forms a complex with a paralogous protein, Yip1; this integral membrane complex is thought to be required either for formation of COPII vesicles at ER exit sites (*Heidtman et al., 2003*) or for fusion of those vesicles with the Golgi (*Barrowmann et al., 2003*). Our genetic data are clearly consistent with a role for Doc in COPII trafficking; for example, the interaction between *doc* alleles and mutants encoding the COPII vesicle components Sar1 and Sec13. Our results are also consistent with a role for Doc/Yif1 in vesicle fusion, given that we did not see an interaction with mutations of the v-SNARE-encoding gene *Syb*, whereas moderate enhancement of the $doc^1$ wing phenotype and semi-lethality with $doc^{99}$ occurred with hypomorphic alleles of the t-SNARE-encoding *Syx 1A*. The enhanced wing-nicking phenotype induced by the $dNSF2^{QE/Q}$ construct we observed in various *doc* mutant backgrounds also suggests a sensitivity to events at the target membrane, given the role of NSF proteins in SNARE complex disassembly (*Hanson & Whiteheart, 2005*).

A notable feature of the Yip superfamily in other systems is that each has several different genes encoding YIP proteins, including four in yeast and nine in mammals (*Shakoori et al., 2003*). The Drosophila Yip superfamily matches that of yeast, with homologs of the Yip1p, Yip4p and Yip5p proteins encoded by the currently unstudied genes *CG12404*, *CG3652* and *CG4645* respectively. The expansion of the family to nine in mammals has functional significance, generating isoforms with distinct, tissue-specific roles. For example, Yif1B is involved in targeting of the serotonin receptor in dendrites, whereas Yif1A is involved in ER-Golgi transport (*Carrel et al., 2008*). Mice homozygous for a null mutation in another family member, *YIPF6*, have recently been shown to have a specific defect in secretion from epithelial cells lining the intestine (*Brandl et al., 2012*). Alternative splicing of the *doc* mRNA produces three different Doc isoforms, differing at their N-terminal end. Although the differences between the Doc A, B and C subtypes are modest, we found that the splice donor and acceptor sites for these alternative splice events are conserved in the *doc* homologs of many other Drosophila species, suggesting that this same set of three Doc proteins are also produced in other members of the genus and are therefore likely to represent functionally distinct isoforms. Furthermore, the location of this variation is highly significant, because the hydrophilic N-terminal end of YIP family members is predicted to project into the cytoplasm (*Matern et al., 2000*; *Yang, Matern & Gallwitz, 1998*), where it interacts with a variety of other proteins such as Rab-family GTPases that play crucial modulatory roles in membrane dynamics

(*Yang, Matern & Gallwitz, 1998*). Isoform-specific antisera or transgenes unable to generate specific subtypes may be needed to reveal any functional differences among the three Doc isoforms.

Of the four *doc* alleles originally identified, and the three that we have examined in detail, $doc^1$ is unique in both its viability in homozygous condition and in producing an oblique wing. Its status as a *doc* allele is confirmed by its rescue using the same transgene that restores viability to the two lethal alleles we tested, so what kind of allele does $doc^1$ represent? Non-lethality and straightforward germline rescue strongly suggests a hypomorph, but none of the lethal *doc* alleles show any visible phenotype when heterozygous, and the semi-dominance of the $doc^1$ oblique wing phenotype also seems at odds with this interpretation. It cannot be hypermorphic either, because increased wild-type dosage from the $doc^+$ genomic construct restores the wildtype wing, rather than making it worse. A third possibility is that $doc^1$ is neomorphic, whereas the lethal alleles are nulls or hypomorphs. This model would be consistent with its semi-dominance and unique wing phenotype, but again this interpretation does not sit easily with straightforward recovery of the wild-type wing by the genomic rescue construct, as well as the qualitatively similar interactions of $doc^1$ and $doc^{16}$ or $doc^{99}$ with $dNSF2^{E/Q}$, *miniature* and Notch pathway components (Figs. 3–5). The ability of the genomic construct to rescue the $doc^1$ oblique wing would suggest that the mutant defect is present within the region of the genome covered by our transgene. However, we have been unable to identify any molecular defects unique to the $doc^1$ chromosome within this region, despite resequencing this genomic interval multiple times. We also could not identify a mutation in the $doc^{99}$ chromosome, but in this case the lethality of the chromosome means there is always a need for a $doc^+$ chromosome present in the sequencing reaction; therefore the wild-type sequence may have hidden a mutant peak in the chromatogram. However, the $doc^1$ mutation is viable and therefore can be sequenced in homozygous condition, so there must be an alternative explanation for this allele. The most likely explanation is that a mutation outside the sequenced region–in a distant enhancer for example–that alters expression of an otherwise wild-type *doc*, in a way that disturbs wing development without affecting viability. However, for this model to explain the $doc^1$ phenotype the rescue construct would have to insert in a genomic location that provided the missing expression in the wing. An alternative and possibly less parsimonious explanation is to invoke epigenetics: perhaps a heritable change in chromatin structure is altering the pattern of *doc* expression and a wild-type copy of the gene in a different epigenetic landscape corrects the problem. This would be an unusual explanation, but such epigenetic modification does occurs in Drosophila, driven by Polycomb and Trithorax group proteins (*Ruden & Lu, 2008*). Sequencing of adjacent sequences beyond the region present in the transgene might reveal additional changes, but confirming that a change on the $doc^1$ chromosome outside the transcribed region is actually causative would not be straightforward. It may be that isolation of new *doc* alleles with oblique wing phenotypes and an identifiable molecular lesion will be required to solve the conundrum of $doc^1$.

## A broad role for Doc in vesicle trafficking

We found that loss of *doc* in the developing wing margin leads to nicks very similar to those produced by some alterations of Notch pathway signaling. Over-expression of Delta was able to restore normal development of the wing blade, indicating that reduced Notch pathway activity was responsible for the marginal nicks; in addition, this result shows that ligand availability is the limiting factor, because Serrate over-expression also produced limited rescue, whereas increased Notch receptor expression did not produce any rescue. The Delta/Serrate rescue of the RNAi-induced nicks also demonstrates that vesicle trafficking pathways are not blocked altogether by expression of the *doc*-RNAi construct, otherwise Delta overexpression would have no effect. This may reflect incomplete knockdown of *doc*$^+$ by the *doc*-RNAi construct, but could also be explained by the presence of a redundant, and Doc-independent, mechanism of Delta transport. It is surprising to us that the *doc*-knockdown-induced nicks could be rescued by increased availability of a single protein, Delta, given that we would expect ER-Golgi transport of a plethora of proteins to be significantly reduced. This might reflect a role for Doc in some, but not all, COPII-mediated ER-Golgi vesicle transport, or perhaps might be due to a very brief window of expression induced by the *C96-GAL4* driver, during which Delta transport is critical. However, our results do confirm that transport of other proteins also depends on Doc, because although Delta overexpression permitted normal development of cells in the wing blade, these otherwise-rescued *doc*-RNAi wings lack many marginal bristles (Fig. 4H). Development of these bristles is known to depend on Wingless-mediated signaling (*Zhang & Carthew, 1998*), so this strongly suggests that another critically important developmental ligand relies on Doc-mediated vesicle transport. We found additional evidence that Doc also plays a key role in earlier developmentally important signaling events, such as the finding that some heterozygous combinations of *N* and *doc* alleles generated synthetic lethality. These results are consistent with a broad role for Doc in vesicle trafficking.

## The link between *doc* and *dumpy*

We originally became interested in *docked* because *doc*$^1$ homozygous flies show an indentation in the distal tip of the wing that is indistinguishable from the phenotype observed in the oblique class of *dumpy* alleles, raising the possibility that the two genes are functionally connected. We have been able to show here, in two distinct ways, that the similarity in phenotype indeed reflects a common mechanism. First of all, there is a strong non-additive interaction between *doc*$^1$ and three different *dpy* alleles, perhaps the most striking example being between *dpy*$^{olvr}$ and *doc*$^1$. Heterozygous animals for the former have wild-type wings, and *doc*$^1$ heterozygotes have only a very slight truncation at the distal tip of the wing; by contrast, animals heterozygous for both *doc*$^1$ and *dpy*$^{olvr}$ show a strong oblique phenotype (Fig. 1). Second, having shown that Doc encodes a protein likely to be involved in ER-Golgi transport we found that reduced dosage of *sar1* and *sec13*, both of which encode elements of COPII vesicles, also produces a marked enhancement of the oblique wing phenotype of *dpy*$^D$. This unambiguously establishes that reduced COPII vesicle trafficking exacerbates *dumpy* oblique mutations, presumably by limiting

transport of Dpy during wing development. Therefore, given that Doc also plays a role in vesicle trafficking, the simplest explanation of the *dumpy*-like wing truncation of *doc¹* is that it results from reduced availability of Dpy. This model has an additional implication: if an oblique wing phenotype can be produced by reducing the concentration of wild-type Dpy in the wing, then the simplest explanation of the *dumpy* oblique alleles is that they are hypomorphs. Also consistent with this model is the sensitivity we observed of Dpy to reduced vesicle trafficking, for example in the much-increased oblique wing we observed in *dpy^D* when dosage of *sar1* or *sec13* was halved (Fig. 5).

The precise lesion in a large collection of *dpy* mutations has been determined, revealing that most alleles are caused by truncating mutations: premature stop codons, deletions or those affecting splice sites (*Carmon et al., 2010*). However, the majority of the non-lethal oblique alleles show a different pattern, not only because the mutations are clustered in a single exon, but also because they are missense. Based on patterns of sequence conservation described in the same study, it was proposed that this exon is alternatively spliced. It may be then that *dpy^o* mutations reduce the availability of a wing-specific isoform, resulting in the observed combination of a homozygous viable, oblique wing phenotype. However, it remains something of a mystery what role might normally be played by such a wing-specific Dumpy isoform, and how a reduced amount of it generates reduced growth in such a specific location. The exon in which the non-lethal oblique mutations occur contains several EGF repeats, so one possibility is that these repeats are modified by Eogt in a way that is necessary for Dpy function during wing morphogenesis.

A puzzling aspect of the *doc-dpy* interaction is why the enhanced *dumpy* oblique phenotypes are only observed with *doc¹*, and not the lethal alleles *doc¹⁶* and *doc⁹⁹*. If the interaction between *doc¹* and *dumpy* is simply from restricted ER-Golgi transport of Dpy, we would expect reduced dosage of the more severe alleles of *doc* to enhance the oblique wing phenotype of *dpy* alleles at least as much as *doc¹*. Part of the answer to this puzzle, as discussed above, probably rests on the unresolved qualitative difference between the *doc¹* and *doc¹⁶/⁹⁹* alleles. However, the range of *doc* allele phenotypes also echoes the behavior of the different classes of *dpy* alleles, which can produce three phenotypes (lethality, thoracic vortices, oblique wings) alone, or in any combination with each other (*Carlson, 1959*). It may be that this is an example of an allele-specific interaction, perhaps due to a neomorphic change in the *doc* expression pattern in the developing wing in *doc¹* that is absent from the other *doc* alleles.

Given that Doc likely functions as a component of the ER-Golgi trafficking machinery, it would be surprising if the interaction we observed between *doc* and *dpy* is unique; unless Doc is only involved in trafficking a limited range of cargoes, interactions with other genes encoding secreted or membrane-bound proteins would be expected. The size of Dpy does not appear to be a critical determinant of the interaction, because we found no hint that mutation of several other genes encoding large cargoes results in a genetic interaction with *doc*. Dpy contains a ZP domain, so we looked for genetic interaction with ten different genes encoding ZP domain proteins; these all share a similar final destination, and may have related roles (*Jovine et al., 2002*) In this case we did find one

gene with a clear interaction: *miniature*. Again, the interaction does not point toward size as being important, because Min is of an unremarkable size, only 682 amino acids long. The interaction extended to multiple alleles of both genes and was non-additive, producing markedly enhanced miniature phenotypes with just a halving of *doc* dosage, even with *doc* alleles that have no wing phenotype on their own. In combination with a small deficiency that uncovers miniature, *Df(1)m259 − 4*, heterozygous combinations of two *doc* alleles with no phenotype alone generated a synthetic lethal when combined. Presumed loss of function *miniature* alleles are homozygous viable (*Roch, Alonso & Akam, 2003*), so this lethality must depend on the combined effect of reduced *m* dosage in the context of reduced transport of other molecules, resulting from the lowered availability of Doc. What then is the common feature underlying the strong *doc-dpy* and *doc-m* interactions? Dpy and Min are both components of the ECM, and it has become clear that, in addition to a variety of structural functions, the ECM is intimately involved in regulating the distribution, activation and delivery of a wide variety of hormones and morphogens (*Hynes, 2009*). This seems to be the case for Min, because recent evidence suggests that a major role for Min in the wing is in regulating diffusion of the insect tanning hormone Bursicon through the ECM (*Bilousov, Kozeretska & Katanaev, 2012*). It may be that reduced dosage of *m*, in combination with a *doc*-dependent reduction in traffic of both Min and other ECM components such as Dpy, disrupts Bursicon signaling enough to produce the enhanced *miniature* wing phenotypes.

Although the precise role of Yif homologs in vesicle transport remains uncertain, the function of the Yip-Yif complex is highly conserved in eukaryotes. This was most dramatically demonstrated in yeast, by rescue of the lethal defect in *Yip1* mutants when transformed with a vector expressing human YIP1A (*Chen & Collins, 2005*). Functional conservation of this family from fungi to mammals suggests very strongly that Doc is likely to play a similar role in *Drosophila*. We have shown here that *doc* encodes the sole Yif homolog Drosophila and anticipate that further study of *docked* may help to clarify the role Yif proteins in vesicle trafficking in eukaryotes.

## ACKNOWLEDGEMENTS

We thank T. Abbeyquaye for his contributions at the origin of this work; undergraduate members of the Thackeray lab who participated in the project; the Bloomington Drosophila Stock Center, the Vienna Drosophila Resource Center and M. Bender for stocks. Flybase was used in a wide variety of ways and was critical to the success of this project.

### Funding

This work was supported by undergraduate and graduate research funds from the Department of Biology, Clark University. The funders had no role in study design, data collection and analysis, decision to publish, or preparation of the manuscript.

## Grant Disclosures

The following grant information was disclosed by the authors:
Department of Biology, Clark University.

## Competing Interests

The authors declare that they have no competing interests.

## Author Contributions

- Suresh Kandasamy conceived and designed the experiments, performed the experiments, analyzed the data, prepared figures and/or tables, authored or reviewed drafts of the paper, and approved the final draft.
- Kiley Couto conceived and designed the experiments, performed the experiments, analyzed the data, prepared figures and/or tables, and approved the final draft.
- Justin Thackeray conceived and designed the experiments, analyzed the data, prepared figures and/or tables, authored or reviewed drafts of the paper, and approved the final draft.

## DNA Deposition

The following information was supplied regarding the deposition of DNA sequences:
The CG5484 genomic sequence in the *doc1* mutant strain is available at GenBank: MT162487.

## Data Availability

The sequence of *doc16* cDNA is available in the Supplementary File. The raw data of the wing measurements represented in Figure 1M is available in the Supplementary File.

## Supplemental Information

Supplemental information for this article can be found online at http://dx.doi.org/10.7717/peerj.12175#supplemental-information.

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
