# Peer review of "A docked mutation phenocopies dumpy oblique alleles via altered vesicle trafficking"

_PeerJ, doi:10.7717/peerj.12175_

## Round 0.1 · original submission · Minor Revisions

Your manuscript has been evaluated by 2 experts in the filed. Although there is enthusiasm about your story, both reviewers list a couple of minor points. Please take all the comments and suggestions into consideration to improve your manuscript.

Reviewer 1 ·

Basic reporting

This paper describes characterization of docked (doc) mutations in Drosophila. The authors found that one viable allele of doc (doc1) phenocopies dumpy oblique alleles. They demonstrated that the doc gene encodes a protein of Yip family, particularly homologous to Yif1 of yeast, which is thought to play a role in vesicular trafficking between the ER and the Golgi. They further examined genetic interaction of doc1 with a variety of mutants involved in COPII transport and showed that there were synthetic effects with sec23, sec13 and sar1 as well as SyxA and dNSF2. From these and other results, the authors proposed that the phenotype of doc1 was due to the reduced transport of Dumpy protein to the extracellular matrix.

Experimental design

It is an interesting paper. Analysis is mostly genetic and the results shown are just morphology of adult wings, but I enjoyed reading it.

Validity of the findings

There are many puzzles remaining, for example, what is the physical nature of doc1 allele and why only a subset of cargoes are affected, but considering that the precise role of Yif1 has not been clarified even in yeast yet, the findings reported here may be helpful for further understanding of the function of Yip family of proteins.

Additional comments

Before deemed acceptable for publication, I would request the authors to improve the following.

1. I am afraid the manuscript was not looked over carefully before submission. Typos and missing references were seen at several places.

2. The discussion is very lengthy. The issues discussed were interesting to me, but many general readers would get tired before reaching the end.

3. In Introduction (line 35), COPII was indicated as coatamer (should read coatomer) protein complex II, but this is not correct (Wikipedia is wrong). Coatomer is the name originally given by Jim Rothman to the complex of subunits of COPI. COP stands for “coat-protein complex.”

4. The authors examined genetic interactions of doc mutations with SNAREs, Synaptobrevin (syb) and Syntaxin (Syx) (line 228 and after), and showed some interactions with SyxA. As there are several distinct sets of SNAREs, the authors should indicate which syntaxin it is. According to the model they propose and also considering the roles of Yip family of proteins in yeast, this particular syntaxin should correspond to the Syntaxin-5 of mammals and Sed5 of yeast. Is it correct?

5. The gene product of dumpy is indicated as Dumpy, Dpy and Dp. It is confusing.

6. The reference Wang et al. (2018) (line 190) cannot be found in the reference list.

7. Typo (line 211): Sec1 -> Sec13

8. What is the ref needed for ZP (line 532).

·

Basic reporting

The paper is very well written and is very clear. There are no major concerns and the minor concerns could be addressed by a re-examination of existing sequencing data and a few straightforward crosses and sequencing reactions.

minor points:

1) Line 190. Although Wang et al 2018 is cited in text, the reference does not appear in the “references” section.

2) Even though the authors state the all images are the same magnification, there should be at least one scale bar for each figure to show the size of the wings.

Experimental design

Generally solid design and well executed.

Minor suggestions:

1) Re the doc[99] allele/chromosome. Unless I missed it, the authors don’t discuss the possibility that doc[99] is a small deletion. They present evidence that the chromosome has both a doc mutation and a closely linked lethal, which supports the hypothesis that doc[99] is a small deletion that takes out doc and at least one adjacent essential gene. This hypothesis could easily be tested in two ways:
A) The authors could test if doc]99] complements amon, the gene two genes to the left of doc, and to lerp, about 100kb to the right.
B) Although the authors state the need for a doc+ chromosome to get DNA for sequencing the doc[99] allele might obscure the presence of a mutation, the need for a doc+ allele might very well reveal the presence of a deletion. Because balancer chromosomes chromosome diverged from common stocks many decades ago, when sequencing heterozygotes such as doc[99]/TM3 there will be polymorphisms visible as overlapping peaks every couple of hundred bases pairs in intronic regions, and reasonably often in wobble bases of coding sequences. If the sequenced DNA amplified from heterozygotes doc[99] lacks double peaks for the doc locus, but shows such peaks for a distant control gene, then doc[99] is a likely a deletion. What might be especially fortunate here is that there is a good chance that doc[16] might have been isolated in the same screen as doc[99] and may have the same parental chromosome as doc[99] and thus could be used to identify polymorphisms to examine in the doc[99]/balancer sequence. The authors are encouraged to look closely at their existing sequencing data for evidence that doc[99] is a deletion.

2) Fig. 1 and Methods. Please briefly describe how is wing length measured. Just the longest line, or as would be more typical, from one landmark to another landmark? If landmarks were used, what landmarks?

Validity of the findings

Conclusions are well supported, alhtoughs some stats are in order for wing length comparisons.

Minor points:

1) Figure 1M and line 140. “ … producing a significant further reduction in wing length (Fig. 1).” The results look highly significant, but appropriate statistical tests should be performed with resulting p-values stated in in text, labeled on Fig. 1M and included in the raw data supplemental table.

Additional comments

This paper identifies the genetically- defined gene “docked (doc)” as being encoded by CG5484, which is the Drosophila homologue of the conserved vesicular transport protein Yif1. Intriguingly, the doc[1] allele causes a wing phenotype that is strikingly similar to the phenotype caused by some mutations in the ECM ZIP domain protein Dpy. The authors characterize several doc alleles for their genetic interactions with the Dpy, as well as mutations in secretory pathway components and other genes trafficked through the secretory system, including components of the Notch signaling system. These analyses provide significant insights in to the function of doc in a multicellular organism.

The presented analysis is broadly interesting to researchers studying secretion, ECM, and organ morphogenesis. The authors do a nice job of considering the role of doc in secretion in context of the interactions with dpy, which is an enormous protein that uses a specialized trafficking pathway, and more conventional proteins such as the notch pathway. Although the authors were scooped in the identification a Drosophila Yif1 homolog (Wang 2018), this in no way detracts from the presented story since the Wang paper focuses entirely on neural functions of Yif1, and does not identify Yif1 as being the gene affected by the doc mutations. Further, there is no consideration in the Wang paper of ECM, wings or the role of doc/Yif1 in trafficking large cargo. Thus this manuscript makes important independent contributions to the understanding of the role of Yif1 in secretion and organ morphogenesis.

---

## Round 0.2 · accepted · Accept

After carefully reading your revised manuscript, the previous comments, and your rebuttal letter, it was concluded that you successfully addressed the reviewer's concerns. Congratulations!